# Comparison of Machine Learning Tree-Based Algorithms to Predict Future Paratuberculosis ELISA Results Using Repeat Milk Tests

**DOI:** 10.3390/ani14071113

**Published:** 2024-04-05

**Authors:** Jamie Imada, Juan Carlos Arango-Sabogal, Cathy Bauman, Steven Roche, David Kelton

**Affiliations:** 1Department of Population Medicine, University of Guelph, Guelph, ON N1G 2W1, Canada; jamiebimada@gmail.com (J.I.); cbauman@uoguelph.ca (C.B.); sroche@acerconsult.ca (S.R.); 2Département de Pathologie et Microbiologie, Faculté de Médecine Vétérinaire, Université de Montréal, Saint-Hyacinthe, QC J2S 2M2, Canada; juan.carlos.arango.sabogal@umontreal.ca; 3ACER Consulting, 100 Stone Rd West #101, Guelph, ON N1G 5L3, Canada

**Keywords:** paratuberculosis, Johne’s disease, disease control, machine learning, random forest, decision tree, diagnostics, dairy farming, cattle

## Abstract

**Simple Summary:**

Johne’s disease is a chronic progressive gastrointestinal disease of ruminants. The diagnosis and control of this disease remains a challenge for cattle producers and herd veterinarians. Machine learning is a broad class of algorithms and statistical analysis belonging to the area of artificial intelligence. These techniques allow for the recognition of patterns in large datasets that are not found by traditional analytical and statistical methods. While artificial intelligence and machine learning have been used in human medicine and in various aspects of animal husbandry, they have only just started to be used in Johne’s disease research. By using results from milk component tests and past Johne’s individual test results, tree-based models, a type of machine learning, were used to predict future Johne’s results. This type of predictive algorithm may help producers and veterinarians more reliably identify those animals most likely to yield a positive result. This will, in turn, help improve control efforts and reduce the testing burden for herds working towards disease control and eradication.

**Abstract:**

Machine learning algorithms have been applied to various animal husbandry and veterinary-related problems; however, its use in Johne’s disease diagnosis and control is still in its infancy. The following proof-of-concept study explores the application of tree-based (decision trees and random forest) algorithms to analyze repeat milk testing data from 1197 Canadian dairy cows and the algorithms’ ability to predict future Johne’s test results. The random forest models using milk component testing results alongside past Johne’s results demonstrated a good predictive performance for a future Johne’s ELISA result with a dichotomous outcome (positive vs. negative). The final random forest model yielded a kappa of 0.626, a roc AUC of 0.915, a sensitivity of 72%, and a specificity of 98%. The positive predictive and negative predictive values were 0.81 and 0.97, respectively. The decision tree models provided an interpretable alternative to the random forest algorithms with a slight decrease in model sensitivity. The results of this research suggest a promising avenue for future targeted Johne’s testing schemes. Further research is needed to validate these techniques in real-world settings and explore their incorporation in prevention and control programs.

## 1. Introduction

Mycobacterium avium subspecies paratuberculosis (MAP) is the causative agent of Johne’s disease (JD). This untreatable, chronic, and progressive disease of ruminants results in economic losses of up to USD 28 million for the Canadian dairy industry annually [1]. Global approaches to on-farm JD control are typically centered around the implementation of management practices that minimize the introduction or within herd spread of MAP and the identification and removal of MAP-positive animals from the herd [2].

The test and cull control method is hampered by the long latency/incubation of the disease before clinical signs, the large proportion of infected populations existing as asymptomatic infectious carriers, and the absence of accurate and reliable diagnostic tests [3,4]. Historically, JD tests have demonstrated a low sensitivity and high specificity. False negatives occur due to the biological characteristics of the disease wherein the shedding of the pathogen and/or the circulation of detectable antibodies occur late in the stage of infection and can be inconsistent over time [5,6]. False positives are much less likely to occur, but, when they do, they can often be attributed to the cross-reactivity of the antigens being identified with molecular tests [6] or due to the “pass-through” phenomenon in the case of organism detection tests [7]. It has also been suggested that animals exposed to environmental non-MAP mycobacterium may generate cross-reacting antibodies, which can produce false positive results for bovine paratuberculosis serologic tests [8]. When tests have an imperfect sensitivity and specificity, producers may have difficulty making culling decisions. It can be difficult to cull a valuable animal which tests positive on a JD test in the absence of clinical signs or frustrating when an animal which tested negative previously demonstrates clinical signs at a later date.

Compounding this frustration is interpreting these imperfect tests over time. Herds which participate in long-term serologic or milk testing frequently encounter conflicting results (i.e., a cow undergoing testing every time they are dried off may have two positive tests followed by a negative test) [9]. Interpreting results that are sequentially inconsistent requires a systematic decision-making process.

To date, this process of interpreting inconsistent sequential results is beyond the ability of researchers with the analytical tools currently available to them. With the advent of machine learning (ML), a subfield of artificial intelligence (AI), the classification of data based on pattern-recognizing algorithms has become possible. The goal of ML is to simplify the data analysis process, increase the reliability of the insight gained, and improve the decision-making process. While ML-guided decision support systems have been used in human medicine since the 1970s, their application in veterinary medicine has been few and far between [10]. Most applications of AI in the veterinary and animal husbandry sector have been focused on solving animal health and welfare problems through analyzing animal behavior [11,12] with very few studies addressing disease detection [11]. In a review by Basran and Appleby [13], the authors highlight the unmet potential for AI in veterinary medicine; however, the published work on AI in veterinary medicine has mainly been centered on medical image analysis (e.g., radiographs, CT, MRI, ultrasound images, and teat images on dairy cattle).

There are many different ML approaches, each with their own unique benefits, depending on the data being used [14]. The ML algorithms are broadly classified as either supervised or unsupervised, depending on whether the data are labeled or not [15]. The supervised ML algorithms, such as decision trees (DTs), random forests (RFs), and neural networks (NNs), are methodologies that can be used to analyze data to help predict or classify future results. Algorithms of the same type may also perform differently. As Matvieiev et al. [16] noted in their research involving artificial neural networks (ANNs) to predict milk yield and milk components, different ANN models were better at predicting different outcomes (i.e., one model that performed well predicting milk yield was not as good as another ANN at predicting protein content).

Researchers have attempted to address various aspects of companion animal medicine using ML and AI, from the evaluation of the criteria in structured histology reports [17] to the differentiation between inflammatory bowel disease and alimentary lymphoma in cats [18] to the early prediction of canine cancer through blood serum analysis [19] and the short- and medium-term survival of acute-on-chronic kidney disease in cats using DT models [20]. In their work using ML to predict leptospirosis, Reagan et al. [21] cautioned that models should have some form of post-implementation evaluation for each new population that the model is applied to. Furthermore, the models developed to predict diseases function as predictive algorithms rather than diagnostic tools, and are best used for identifying animals at higher risk, who should thus be prioritized for further analysis [22,23].

Much of the AI and ML research in dairy farming has focused on behavior classification, early lameness, estrus detection, fertility predictions, and disease detection (mostly subclinical mastitis and ketosis) [24]. In their review of 142 publications covering ML in dairy farming, Lokhorst et al. [25] found that, among the 134 studies that used supervised learning, 64% used it for classification. This finding is also reported by Slob et al. [26] and Wang et al. [27], who found that decision tree-based algorithms, a type of classification approach, are by far the most utilized followed by ANNs and support vector machines (SVMs).

Schmeling et al. [28] used the data from accelerometers, magnetometers, and gyroscopes to predict cow lying behavior. The researchers used RF, decision tree, SVM and naive Bayes to model the data with the RF achieving the highest accuracy. Zhou et al. [29] utilized collar data alongside milking system data with eight different ML algorithms to predict common disorders in dairy cattle with SVM, DT, and RF performing the best (accuracy >80%). The benefit that these ML methods have in dairy disease predictions (and disease detection in general) is that they can discover complex, latent patterns between predictor variables and the trait of interest (the disease status) even if this relationship is non-linear. These methods do not need to have a preconceived hypothesis compared with other more traditional statistical methods as their purpose can be solely to identify patterns [30]. However, a major hurdle in the use of AI and ML on dairy farms is that dairy-related data exists in silos that are not connected [24].

Another challenge with some machine learning algorithms is their lack of interpretability. To leverage the power of machine learning methods while avoiding this lack of interpretability, Sykes et al. [31] used explainable AI models to assess the impact of on-farm biosecurity practices on the predicted risk of a Porcine Reproductive and Respiratory Syndrome (PRRSV) outbreak. The researchers utilized SVMs, RFs, and gradient boosted machines in their study and performed a variable importance analysis to identify the most influential factors within the models. The generated models have the potential to inform individualized biosecurity assessments, informing which practices are most important at a specific farm.

While the current research with ML being applied to disease surveillance and risk assessment has been limited, ML does show promise as a useful tool. Pereira et al. [32] used 77 predictors related to the type of farm, type of lactation, number of animals on the property, and gradient boosted trees to predict which farms were at the highest risk for tuberculosis infection. Stański et al. [33] used various ML techniques such as NNs, RFs, gradient boosted trees, and support vector classifiers to improve the predictions of herd-level bovine tuberculosis beyond that of the traditional herd-level single intra-dermal comparative cervical tuberculin test. By using data on the farm location, animal movements, links to infected farms, past testing history, and land cover around the farm, the researchers were able to demonstrate an increase in herd sensitivity from 61 to 67% and herd specificity from 90 to 92%.

Various groups have attempted to use ML methods to evaluate, classify, and predict udder and teat health. Researchers have used data such as milking parameters [34], cytometric fingerprints [35], and SCC [36] to predict udder health and mastitis. Moving beyond classifying the health status of the udder, Heald et al. [37] used ANNs to classify the bacteriologic causes of mastitis. Others have attempted to utilize image data, such as Porter et al. [38] who used convolutional neural networks (CNNs) to classify images of teat ends achieving an AUC of 77%. The authors noted that subtle differences between some classes may have limited the classification performance of their model. Researchers have also utilized the variable importance of ML models to identify those factors which are the best predictors of disease [39,40] or, in the case of Johne’s research, to identify the most influential control measures for disease prevention [41].

Machine learning is a family of statistical analysis techniques that are relatively new to MAP research. Volatile organic compounds (VOCs) have been explored as a possible marker for infection with MAP [42,43], demonstrating the possible potential for detecting infection prior to bacterial culture results [42]. Meanwhile, others have attempted to utilize gut microbiota composition to predict the infection status and shedding severity [44,45]. Johne’s ML research continues to expand to explore new avenues in the detection of MAP. The changes to sera microRNA [46] and metabolomic changes in polyunsaturated fatty acids and eicosanoids [47] have been used to differentiate between MAP infected and non-infected animals.

Researchers have highlighted that there are still challenges with the application of ML. Obtaining quality data from multiple sources remains one of the biggest barriers [48,49]. Furthermore, the datasets need to be accurately labelled for supervised learning techniques to be effective [50]. As in most modeling, the quality of the model is dependent on the quality and representativeness of the datasets used by the ML algorithms [48,51,52]. In their review of the applications of ML in animal and veterinary public health surveillance, Guitian et al. [51] note that ML algorithms can and have been used to increase the likelihood of pathogen detection, allowing for the prioritization of samples most likely to yield a positive result. However, few studies have explored the efficient and practical deployment of ML algorithms to real-world scenarios [52,53]. Hennessey et al. [49] concluded that the establishment of large, labeled, open-source data sets and the incentivization of ML researchers to take an interest in the veterinary space will be essential for progress to continue, as it has in human medicine.

## 2. Objective and Hypothesis

The main objective of this study was to assess the ability of tree-based classification models to predict a Johne’s test result based on concurrent milk components testing results, cow demographic data, historical Johne’s testing of the cow, and the Johne’s positivity rate of the herd. As a secondary objective, we compared the results of decision tree (DT) and random forest (RF) models to illustrate the trade-off between complexity and interpretability commonly faced when fitting ML models. We hypothesize that tree-based classification models can incorporate milk component test results with past Johne’s test results to provide accurate predictions on future test results. Further, DTs might have a comparable performance to RFs with results that are more easily interpreted by stakeholders.

## 3. Material and Methods

### 3.1. Data Preparation

The data for the current project were obtained from a longitudinal study conducted on Ontario dairy farms from 2010 to 2014. The farms enrolled in this study were conducting frequent (2–4 times per year) whole herd individual cow JD testing and were offered financial compensation for the tests in exchange for access to the test results. These cow-level results consisted of 4407 IDEXX milk ELISA (Idexx laboratories, Westbrook, ME, USA) or Parachek^®^ milk ELISA (Prionics USA, Inc., Omaha, NE, USA) results and their corresponding interpretation based on the manufacturer’s instructions (interpretation) (i.e., positive, negative, or suspect) with an additional interpretation category (i.e., high) that represented strong positive ELISA results (>1.0 optical density) [54] for 1197 cows. This “high” classification was part of the Ontario Johne’s control program “http://www.johnes.ca/” (accessed 2 November 2021), with the rationale being to target and cull “highly positive” animals to remove animals shedding high amounts of pathogen. All herds in this study were enrolled in milk recording through Lactanet, and the JD tests were conducted on samples collected as part of the herd’s routine milk testing. The demographic data and milk testing results—lactation number (lact), cow breed (breed), days in milk (dim), 24 h milk yield (m24), 24 h fat yield (f24), 24 h protein yield (p24), predicted 305-day milk yield (m305), predicted 305-day protein yield (p305), predicted 305-day fat yield (f305), somatic cell count (scc), somatic cell count linear score (sccls), and milk urea nitrogen (mun)—from the corresponding test dates were retrieved from the dairy herd improvement records as collected by Lactanet (*n* = 4750). The ELISA results from the longitudinal study were combined with the milk testing data and matched by test day and animal identification number (*n* = 4008). The 4008 JD ELISA results with matched milk testing data represented the results from 1059 animals (70% Holstein, 30% Jersey, average scc = 253.72, average m305 = 8372) across eight different herds, of which there were 164 cows with only one Johne’s test result. A total of 895 cows had a minimum of two JD test results. Seven of these animals had a maximum of 10 test results (Figure 1).

While over 84% of the cows were tested for JD multiple times, each JD test result was considered as an independent observation in the dataset (*n* = 4008), because each result represented an animal in different lactations, stages of lactation, and involved different protein yields, fat levels, and milk yields, so accommodation was not made for within cow correlation. For each observation, additional variables were created to capture previous Johne’s test results for that particular animal (i.e., the number of previous tests (prevtests); the number of tests including the current test that were negative (neg), positive (pos), suspect (susp), or high (high); the number of previous tests that were negative (prevneg), suspect (prevsusp), or high (prevhigh); the proportion of previous tests that were positive (proppos); and the average ELISA (avELISA) score of previous tests). Both the proppos and avELISA were created from all test results except the most current result. The proppos was created by dividing the number of previous positive tests by the total number of previous tests. The avELISA was created by calculating the average ELISA OD from the previous JD tests. In addition, two variables (herd.pos.rt and herd.pos.rt.and.susp) were created that described the context of the herd Johne’s positivity in which the animal originated (i.e., the herd positivity rate). This herd positivity variable was created using the whole herd results for the specific test day, with the number of positive results on test day divided by the total number of animals tested. The herd positivity rate was defined as low (L) if less than 5% of the herd tested positive, medium (M) if 5–15% of the herd tested positive, and high (H) if greater than 15% of the herd tested positive on a specific test date. This herd positivity rate was calculated using those results that were reported as positive (positive + high) (herd.pos.rt) and calculated again with the inclusion of suspects in the numerator (suspects + positives + highs) (herd.pos.rt.and.susp).

The data processing and analysis were performed using R Studio Version 1.3.1093. The variables with few or no observations were identified through subsetting and plotting using the heatmap function of the stats package. Variables missing more than 3500 or 80% of their observations were excluded from further analysis. Due to the nature of the RF models, the correlation between the variables may not significantly affect the model outcomes, due to the fact that variables are selected at random for each tree of the RF. However, the remaining variables (after removal of those variables missing 80% of their observations) were checked for collinearity using the Spearman rank correlation coefficient and the cor function in the stats package. The highly correlated variables were identified as those with values greater than 0.8. Of the highly correlated pairs of variables (>0.8), if the variables represented a similar biological measurement, the variable with the most relevance was retained in the model, otherwise the highly correlated pairs of variables both remained in the models. The categorical variables were evaluated for correlation through the predictive power score function pps from the ppsr package.

After removal of the highly correlated variables representing a similar biological measurement and those missing large amounts of data, four different data subsets were generated from the full dataset to test the RF model with different variables. For the first and third data subsets, the JD test results (e.g., propos, avELISA, prevtests, etc.) were removed as predictors to assess how milk components performed as predictors on their own. Within the JD-related predictors, the variables avELISA and proppos were only applicable to animals with repeat tests; therefore, the second and fourth subsets consisted of only observations that contained at least one previous test result. This was achieved by removing the first test for each animal (as the animal at this test had no previous results), as well as animals with only one test. The third and fourth subsets were created to address the concern that there were only 257 ‘positive’, 49 ‘high’, and 37 ‘suspect’ results, as well as 3665 ‘negative’ results, and the low number of high and suspect classifications may limit the prediction accuracy for the model. Therefore, for the third and fourth subsets, the outcome (interpretation) of the first and second subsets was changed from multiclass (negative, suspect, positive, and high) to dichotomous (negative and positive) with the ‘high’ and ‘positive’ results being classified in the positive category and the ‘suspect’ and ‘negative’ results pooled in the negative category (Table 1).

### 3.2. Model Building

Training and testing datasets were created by splitting each of the four data subsets 3:1 (training: testing) using the initial_split function in the rsample package. Due to the fact that the ELISA interpretation results were unbalanced in the data subsets (with many more negative values than positive, suspect, or high results), the data split was stratified by interpretation so that both the training and testing datasets had a similar proportion of positive, suspect, high, and negative observations (Table 2). Using the tidymodels package, the baseline versions of the RF models were trained using the training dataset from each data subset, through bootstrap aggregation, with a default of 500 trees and a random selection of the predictor variables for each node.

The models were assessed using accuracy measures (accuracy and kappa) and the Variable Importance Plots (VIP) function (Table 3). In the case of accuracy, it is the proportion of data that is correctly predicted, whereas kappa is the proportion of data that is correctly predicted while adjusting for the agreement expected from chance alone. Once both the training and testing were completed on the basic model, three hyperparameters (the number of trees [tree], the minimum number of observations to split at any node [min_n], and the number of variables sampled at each split [mtry]) were utilized for tuning using the tuning function to achieve the highest possible accuracy. This was completed through the creation of a tuning grid and testing the various hyperparameters on five separate cross-validated folds of the training data.

Using the combination of hyperparameters with the highest accuracy (kappa) obtained in the cross-fold validation of each training dataset, the final RF models for each subset were trained using the training dataset, and predictions were generated using the test dataset. The models were again analyzed using the VIP function. The performance of the model was assessed through accuracy measures (kappa and roc area under the curve (AUC)), out-of-bag error, plotting gain curve, and a confusion matrix. Gain curves, also known as cumulative gain curves, are a method for evaluating the performance of a classification model. They demonstrate the percentage of observations in a given category when considering a certain percentage of the observations with the highest probability to be the target category (according to the model). The assessment of overfitting was performed through comparison of the kappa of the trained model and the kappa of the model applied to the test dataset.

The most accurate model generated from the four data subsets was further analyzed through a confusion matrix to describe the model’s sensitivity, specificity, positive predictive value, and negative predictive value. Additionally, an exploration of the hyperparameter combinations was completed on the model to demonstrate the effects on kappa.

Decision tree (DT) models were then built using the data subset of the most accurate RF model and the rpart function of the rpart package to assess if a gain in interpretability can be achieved without compromising accuracy when compared to the best performing RF model. The DT was first built with a training subset and the performance was assessed on the testing subset of data. Similar to the approach taken with the RF models, the DT hyperparameters (cost_complexity and tree_depth) were utilized for tuning using the tuning function to achieve the highest possible accuracy. This was completed through the creation of a tuning grid and testing the various hyperparameter combinations on five separate cross-validated folds of the training data. The performance (accuracy, kappa, sensitivity, specificity, positive predictive value, negative predictive value, AUC, and gain curve) of the DT was also evaluated.

## 4. Results

Among the 895 animals with more than one test result, 92 had a different interpretation result for their last test compared to the result of their first test. There were five animals that changed from positive on their first test to negative on their last test. In total, 87 changed from negative on their first test to positive on their last. Of the 803 animals with an unchanged test result for their last test compared to their first test result (770 negative cows, 33 positive cows), 13 had at least one intervening test that was opposite to the first and last test (e.g., a cow that tests negative on first test, positive on second test, and negative on third test). The variable scc was removed due to a high correlation with sccls and to simplify the model, while mun was dropped due to a high number of missing values (*n* = 3507).

### 4.1. Model 1

The first model was trained and tested on data subset 1 (*n* = 4008 observations).

The data contained 10 predictor variables representing the results for milk component testing and an outcome variable with four levels (negative, suspect, positive, and high). In the baseline model with 500 trees, the most important variable in the model was m305. This model yielded an accuracy of 0.913 and a kappa of 0.021. Tuning the hyperparameters (mtry, trees, and min_n) for kappa yielded a model with mtry = 5, trees = 1, and min_n = 2. Over the 5-fold cross-validation, this combination of hyperparameters averaged a kappa of 0.075 and an out-of-bag error of 0.162. The variable m305 remained the most important variable in the optimized final model for subset 1. When the model was applied to the test dataset, a kappa of 0.054 and a roc AUC of 0.514 was achieved (Table 1). There was evidence of overfitting as the model performed better on the trained data (kappa = 0.075) than on the test data (kappa = 0.054). Plotting the gain curve for the model demonstrated a poor predictive ability across all four levels of the outcome (interpretation) (Figure 2).

### 4.2. Model 2

The second model was trained and tested on data subset 2. This subset consisted of 2949 observations with 18 predictor variables and an outcome variable with four levels (negative, suspect, positive, and high). The most important variable for this baseline model (500 trees) was avELISA. The model yielded an accuracy of 0.924 and a kappa of 0.508. Optimizing the hyperparameters of the model for kappa generated a model with mtry = 18, trees = 1000, min_n = 21, and an out-of-bag error of 0.063. The average kappa across five folds of the training data was 0.543. The variable propPOS was the most important variable within the model with the optimized hyperparameters. When the model was applied to the test dataset, a kappa of 0.518 and a roc auc of 0.763 was achieved (Table 1). There was no clear evidence of overfitting as the model performed only marginally better on the trained data (kappa = 0.543) than on the test data (kappa = 0.518). Plotting the gain curve for the model demonstrated a better performance across all outcomes than with the model generated from just the milk component testing results (Figure 3).

### 4.3. Model 3

The data subset used for model 3 contained the same number of observations (and predictors) as data subset 1; however, the outcome variable was dichotomous (positive, negative). The most important variable for the baseline model 3 was m305, similar to the baseline model 1. Applying this baseline model 3 to the test data demonstrated an accuracy of 0.915 and a kappa of 0.019. The hyperparameters optimized for kappa were mtry = 1, trees = 1, and min_n = 21 with an average kappa of 0.076 and an out-of-bag error of 0.084. The most important variable in this optimized model was f305. Applied to the test dataset, the model had a kappa of 0.038 and a roc AUC of 0.372 (Table 1). As in model 1, there was evidence of overfitting as the model performed better on the trained data (kappa = 0.076) than on the test data (kappa = 0.038). The gain curve (Figure 4) demonstrated that less than 50% of the positive tests within the testing dataset are in the top 25% of observations with the highest predicted probability of testing positive.

### 4.4. Model 4

The final model utilized data subset 4, which contained the same 2949 observations as data subset 2, with 16 predictor variables and a dichotomous outcome variable (positive, negative). The most important variable in baseline model 4 was the avELISA, and the model achieved an accuracy of 0.943 and a kappa of 0.614 when applied to the training data. The optimal (for kappa) combination of hyperparameters for model 4 was mtry = 8, trees = 2000, and min_n = 2, with an out-of-bag error of 0.038. This combination of hyperparameters achieved a kappa of 0.616 across five folds of the training dataset. Similar to the baseline model, the avELISA was the most important predictor in the model. When applied to the test data, the optimized model 4 demonstrated a kappa of 0.626 and a roc AUC of 0.915 (Table 1). Similar to what was seen in model 2, the performance of model 4 applied to the training data was similar to that of test data, and therefore no evidence of overfitting was present. Through plotting the gain curve (Figure 5), it was seen that over 80% of the positive test results were within the top 25% of the observations with the highest predicted probability of testing positive. Finally, a confusion matrix was tabulated with the predicted classes vs. true classifications. From this, the sensitivity and specificity of the model was calculated with a sensitivity of 72% and a specificity of 98%. There was a positive predictive value of 0.81 and a negative predictive value of 0.97.

To further explore the changes in the kappa associated with different combinations of hyperparameters, six separate RF models were trained and tested on data subset 4 without the use of a tuning grid (Table 4). Model 4-D (mtry = 8, min_n = 2, and trees = 1000) yielded a kappa of 73.7% and an AUC of 0.947 (roc curve in Figure 6). In comparison to model 4-A (mtry = 8, tree = 2000, and min_n = 2) that correctly predicted 84% of positive tests in the top 25% of test results with the highest predicted probability, model 4-D was able to correctly predict approximately 90% of the positive tests. The confusion matrix of model 4-D yielded the same sensitivity (72%), specificity (98%), positive predictive value (0.81) and negative predictive value (0.97) as model 4-A.

### 4.5. Decision Tree

A DT was built using data from subset 4 as this was the same data used in the best-performing RF model. As seen in Figure 7, the most important variable within the tree was the avELISA. According to the DT, less than 3% of animals with an avELISA less than 0.046 were classified as positive. However, 92% of animals with an avELISA equal or greater than 0.11 and more than one third of their previous tests being positive were classified as positive (Figure 7). The DT achieved an accuracy of 0.951, a kappa of 0.629, a sensitivity of 53%, a specificity of 99%, a positive predictive value of 0.85, and a negative predictive value of 0.96. The gain curve of the DT (Figure 8) demonstrates that less than 70% of the positive test results were within the top 25% of the observations with the highest predicted probability of testing positive. Figure 9 provides the output of the receiver operator curve which resulted in an AUC of 0.803. When the DT hyperparameters were optimized for kappa, an AUC of 0.780 was achieved. This DT contained only one split, avELISA < 0.046.

## 5. Discussion

The tree-based models evaluated in this study (models 2 and 4) accurately predicted JD milk test results. As has been previously described [55,56], the uncertainty of JD test results, due to poor diagnostic performance, makes management decisions based on these results challenging. There were 105 animals in our dataset that had results that changed over the course of their testing history. These animals represented 12% of all animals with multiple tests; however, they illustrate an important issue in terms of confidently selecting animals to be potentially removed from the herd to limit the possible spread of JD.

Model 1 and model 3 showed evidence of overfitting; however, there was no evidence of overfitting in model 4 as the kappa on the testing dataset was higher than that achieved on the training data, and the kappa for model 2 applied to the test data was only slightly lower than the kappa on the training data [57]. This observation could be due to both the data subsets 1 and 3 containing too few observations for the number of predictor variables. Efforts were made to minimize overfitting by using cross-fold validation of the training data for the tuning of the hyperparameters and bootstrap aggregation for training the optimized models. The number of variables tried at each split was low to help promote randomness within the models.

The model including the concurrent milk component testing results, cow demographic data, historical JD testing of the cow, and JD’s positivity rate of the herd (model 4 with 16 predictor variables with a dichotomous outcome) showed the highest accuracy in its predictive probability. This model may provide a tool to help producers target cows for follow-up testing. For example, the final RF model 4-D was able to capture 90% of the positive tests in the top 23% of its predicted classifications. This means that focusing on the results of 173 tests out of 738 total testing dataset results would identify 64 of the 71 positive tests. This suggests that a strategy based on the results of this model could potentially replace whole herd testing as a screening method, by only targeting those animals most likely to generate a positive result. This targeted or weighted testing strategy would, in effect, more efficiently identify candidates for removal from the herd.

The strategy would be based on the predictions generated from the 16 predictor variables included in the model (the most influential variables for model 4 can be found in Figure 10). As seen in the analysis, the variables avELISA, representing the average ELISA OD of previous tests, and proppos, or the proportion of previous tests that were positive, were the most influential variables in the final model 4 (Figure 10). This finding is not surprising as the more consecutive positive results from previous tests from an animal, the more confident we are that the most recent positive ELISA is not a spurious result. Other researchers have demonstrated that individual animals with high ELISA S/P ratios on previous tests were more likely to test positive on their next test [9,55].

It should be noted that an important input of this model is previous Johne’s test results, and therefore it is likely that whole herd testing results would still be required to implement this methodology into a control program. However, in the case of whole herd repeat testing, one round could be reserved for this targeted approach (i.e., a producer doing quarterly whole herd testing could elect to use a targeted approach for one or two of these quarterly tests).

As seen in Table 3, out of the 18 variables that were used to create the RF models, the milk protein and fat concentrations, milk yield, DIM, and sccls were the predictors that were within the 10 most important factors across the four final RF models. The lactation number and herd positivity rates were also important predictors in the RF models. Unfortunately, the interpretation of the variables, beyond their importance in predicting the outcome, is difficult with RF algorithms, often referred to as ‘black box’ algorithms [58]. However, we can see that the variables that had a large influence in our models’ predictions were also significant in other research studies. The association between JD antibody milk positivity and DIM has been demonstrated by Nielsen et al. [59], with cows at the beginning of their lactation being more likely to test positive. Similarly, Nielsen and Toft [60] found that DIM and milk yield had a significant effect on the odds an animal would test positive on milk ELISA.

Fitting a DT on the same data allows for an interpretable model (Figure 7) at the cost of prediction accuracy and model performance (Figure 8 and Figure 9). This result is not unexpected as random forest models are generally more accurate than individual decision trees because they are an ensemble classifier utilizing multiple trees. This classification results in less overfitting, which provides a better predictive performance [14]. Pruning the DT through hyperparameter optimization resulted in a DT with one node, which results in a much simpler approach to classification with the drawback of having fewer “pure” leaves (e.g., in the group of animals classified as positive, there will be a higher proportion of animals misclassified). Using the DT approach allows a more “hands-on” option for the producer or veterinarian to make decisions surrounding culling or further testing for Johne’s management. In comparison, the use of the RF allows for better sensitivity; however, it requires more technical expertise and does not allow for a directly interpretable approach to classification.

It is important to consider that the models generated here are predicting a positive test result of an imperfect test; therefore, it may be possible that some cows predicted as positive are not. However, when considering the most important variables used in predicting the positive result, it is most likely that they are MAP-infected. The ML models used in this study were mainly focused on kappa as the main measure of model performance and did not rely heavily on accuracy. Previous work has shown that accuracy was often an overestimate of model performance when there was a disproportionate number of negative results [57]. Kappa also appears to be a better performance measure for multi-class classification since it allows for the appropriate assessment of how many overall results were classified correctly above chance alone [57].

The classification task may also be more challenging with an increase in classes. As Viazzi et al. [61] demonstrated, a multi-class classification model did not perform as well as their dichotomous classification model. One of the challenges with RF models is that the training and testing of the models requires an adequately large amount of data. As seen in the comparison of the performance of the model using dichotomous data vs. the model trained and tested on data with four outcomes, the sparsity of results with suspect and high cases results in a poor overall performance of the model (AUC of 0.763 for model 2 with four levels of outcome vs. AUC of 0.915 for model 4 with two levels of outcome). There were over 139,000 cows tested over a 4-year period as part of the Ontario Johne’s Education and Management Assistance Plan “http://johnes.ca” (accessed on 2 November 2021). The Ontario Animal Health Network reported that 3509 ELISA tests were performed per year in Ontario alone (2021 Q3 OAHN Bovine Veterinary Report, “https://www.oahn.ca” (accessed on 23 February 2022)). This illustrates the potential opportunity to explore machine learning if more detailed data could be collected with these tests.

A possible limitation of the above study was the assumption that the observations used for the RF models were considered independent. While the assumption of independence is not required to build RF models, the lack of independence in our dataset may result in a skewed prediction accuracy. Dependence at the animal level as well as at the herd level could potentially create a model that is overfitted to the data. Another limitation in the current dataset was that while the ELISA results were available in both an interpreted result (i.e., positive and negative) and as an optical density (OD), two different commercial tests had been used. The two tests did not have the same classifications, and thus OD considered on one scale may have led to misclassifications on certain observations, particularly those with a low positive OD. The presented modelling approach should be reevaluated when we have data that are based on the results of one test.

While the dataset initially contained many variables to use as potential predictors, a number of variables were not viable in some of the models and adjustments were required to the datasets and models. For example, the values for the variables proppos and avELISA could not be calculated for animals with no previous test. The variables neg, pos, susp, and high captured the total number of negative, positive, suspect, and high tests, respectively; however, this variable also counted the outcome of the current result. This resulted in a falsely elevated prediction accuracy, and these variables were subsequently excluded. Wagner et al. [62] found that the day-to-day and animal-to-animal variability of activity limited the predictive performance of their models (predicting subacute ruminal acidosis), emphasizing that the nature of the predictors being used in the models must be considered. This importance of predictor selection is echoed by Naqvi et al. [63], where simulated models demonstrated that data quality and within data variability impacts the model performance. This suggests that model performance may be limited if predictors are only chosen from existing or routinely collected data rather than features/predictors selected a priori based on prior knowledge (based on the associations between outcome and predictors). Researchers designing studies to develop deep learning models should therefore ensure that predictors are selected in such a way as to reduce as much as possible the unexplained farm- and cow-level variability to maximize model performance, whereas the between-farm differences in disease recording and sensor availability seem to have a smaller impact on the performance [63].

## 6. Conclusions

Machine learning is a tool that shows great promise in helping utilize patterns in data to predict outcomes. Its application in dairy science, dairy cow management, and veterinary medicine is still in its infancy. In our study, RF models provided a higher accuracy compared to DT models to predict JD milk ELISA positive results. However, the latter had the benefit of being easily interpreted by end users having to make culling decisions. Very few paratuberculosis studies have utilized these methods. A challenge with these approaches is understanding exactly how best to apply them to certain datasets. Random forest models allow us to incorporate measurements that may not be clearly associated with our outcome of interest but lend important information to predictive modelling. The RF models generated in our study show the potential to reduce the burden of testing for paratuberculosis. The use of milk testing data was of some importance to the models but was only of minor importance when in the presence of prior JD test data. Ideally, researchers should plan their future studies so that the data collected can accommodate ML analyses. The collection of large amounts of data, both as independent results as well as multiple variables per result, will assist us in generating better predictive models. Future opportunities exist with the collection and analysis of more detailed data, specifically, a standardized ELISA OD, the trend data for milk components, metabolites such as BHB and MUN, and the use of average ELISA OD as predictors.

## Figures and Tables

**Figure 1 animals-14-01113-f001:**
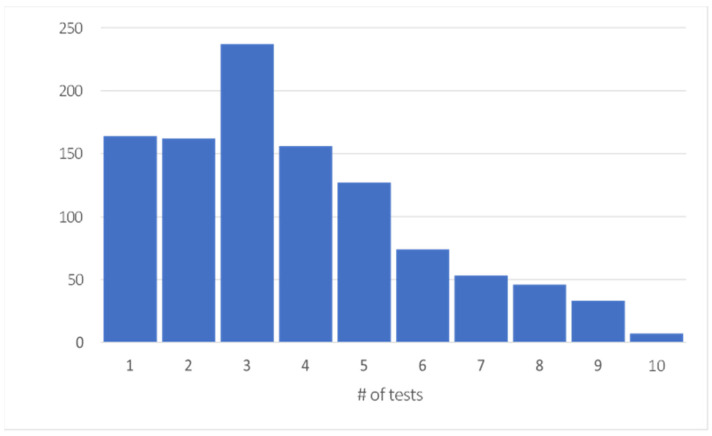
Distribution of the number of tests per cow for 1059 dairy cows undergoing serial Johne’s disease milk ELISA testing on eight Ontario farms.

**Figure 2 animals-14-01113-f002:**
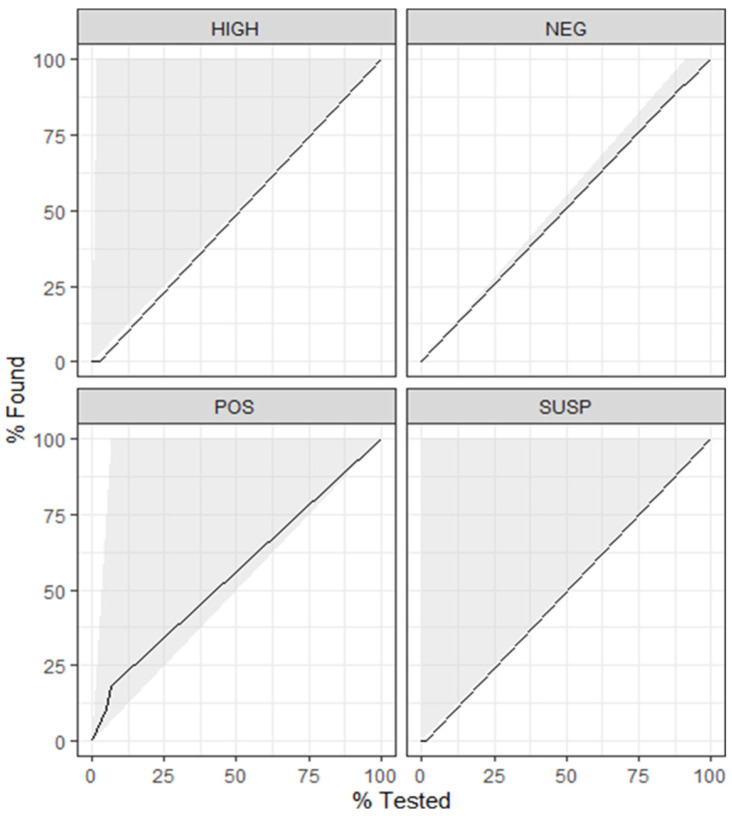
Gain curve for random forest model 1 using the milk component testing results and demographic data to predict the Johne’s milk ELISA result across four levels. The grey shading represents the perfect gain curve. Model 1 demonstrates a poor predictive accuracy across all four levels of the outcome.

**Figure 3 animals-14-01113-f003:**
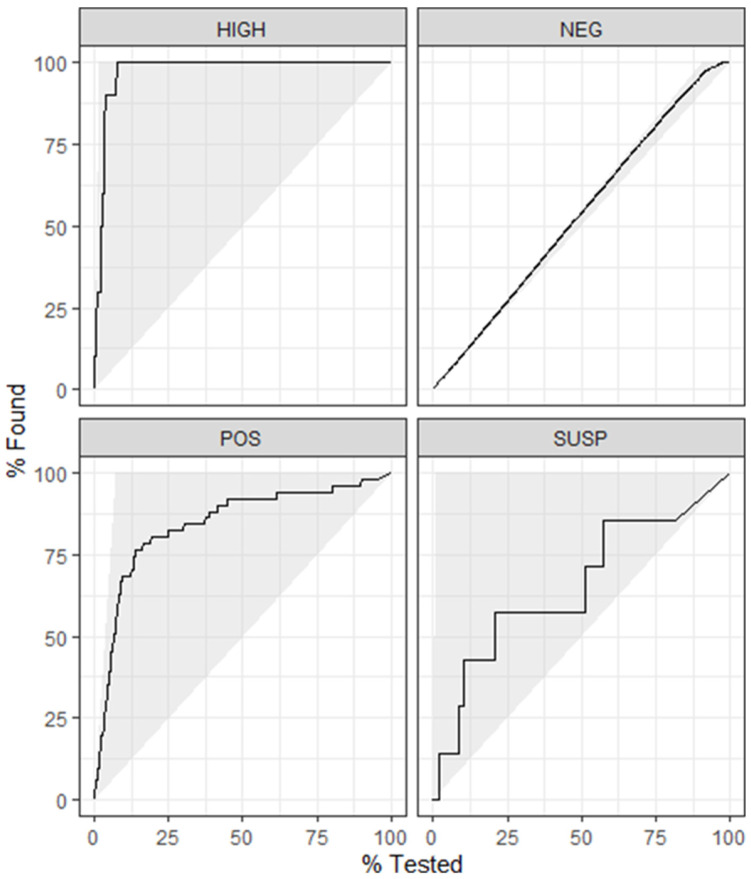
Gain curve for random forest model 2 using the milk component testing results, demographic data, and Johne’s test results to predict the Johne’s milk ELISA result across four levels. Model 2 demonstrates an improved prediction accuracy with the addition of Johne’s test results; however, the model performs poorly for ‘suspect’ test results.

**Figure 4 animals-14-01113-f004:**
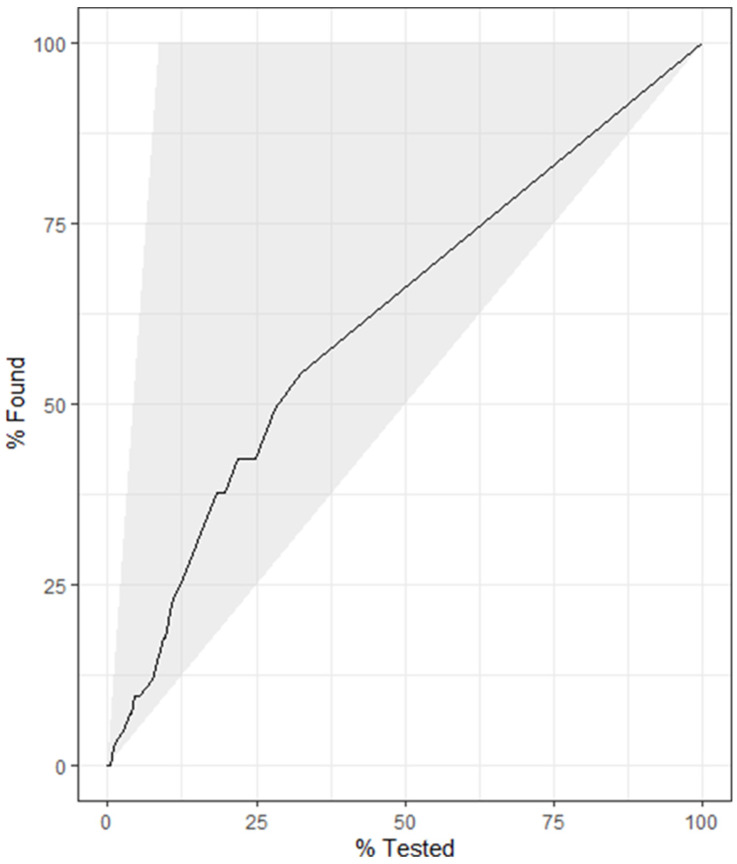
Gain curve for random forest model 3 using the milk component testing results and demographic data to predict the positive test results (Johne’s milk ELISA). Less than 50% of the positive test results were correctly predicted within the top 25% of the observations with the highest predicted probability of testing positive.

**Figure 5 animals-14-01113-f005:**
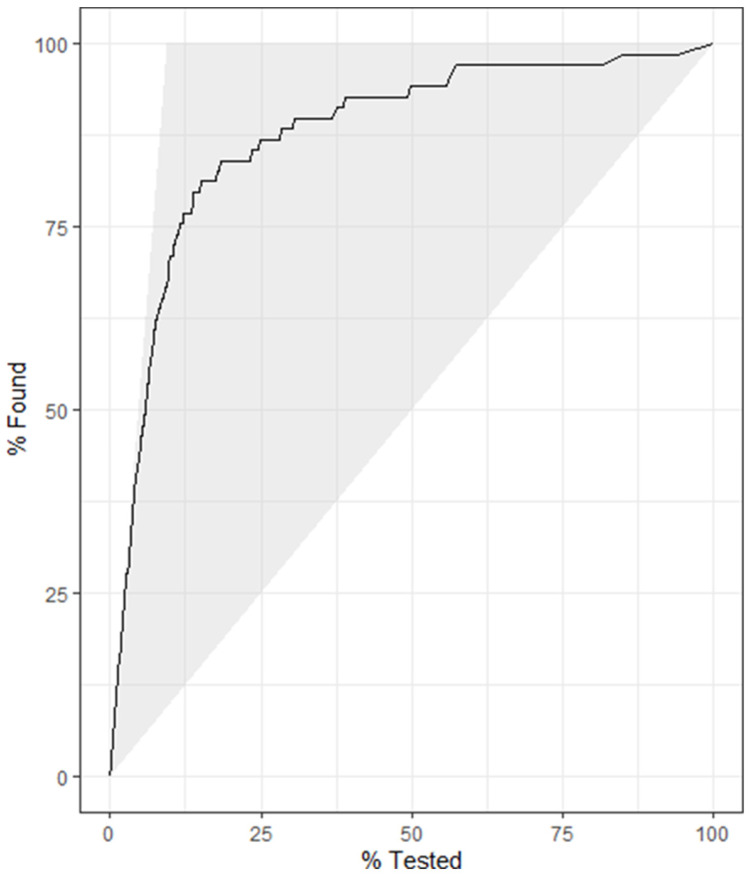
Gain curve for random forest model 4 using the milk component testing results, demographic data, and Johne’s test results to predict the positive test results (Johne’s milk ELISA). Over 80% of the positive results were correctly predicted to be positive in the model’s top 25% of the observations with the highest predicted probability of testing positive.

**Figure 6 animals-14-01113-f006:**
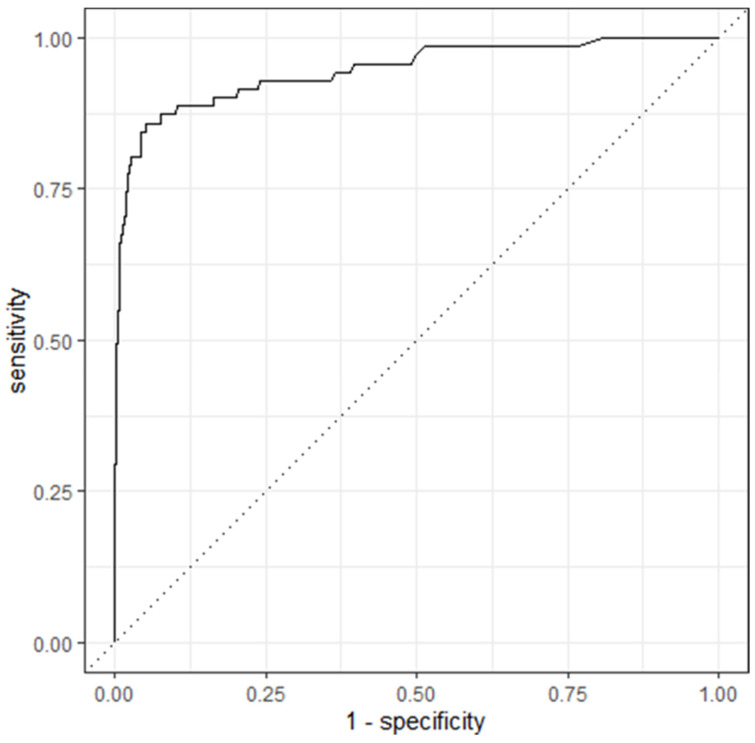
Receiver operating characteristic curve for model 4-D (random forest model predicting the result of the Johne’s milk ELISA) with an area under the curve of 0.947. Model 4-D = random forest model with the following hyperparameters: mtry = 8, trees = 1000, and min_n = 2. Mtry = number of variables sampled at each split, trees = number of decision trees, and min_n = the minimum number of data points in a node that is required for the node to be split further.

**Figure 7 animals-14-01113-f007:**
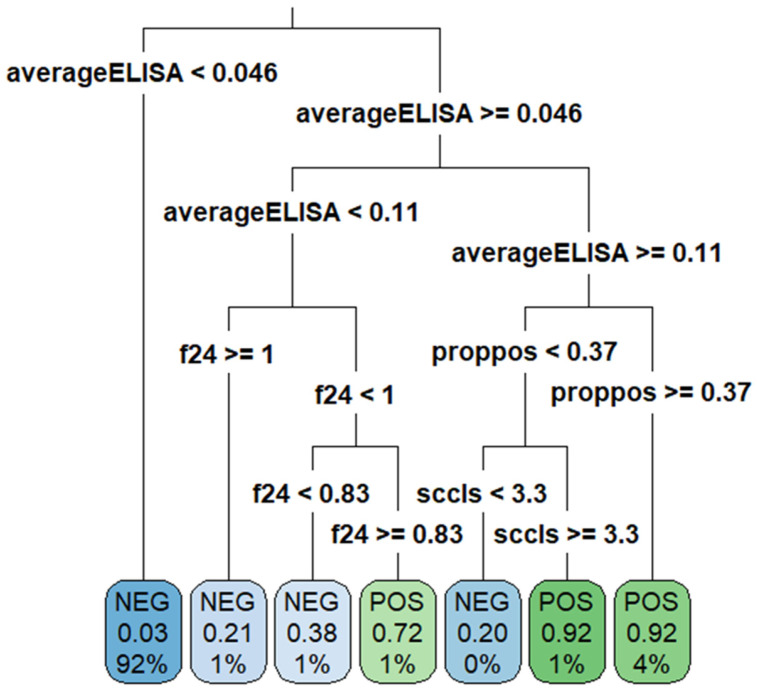
Decision tree to predict test results (Johne’s milk ELISA). The bottom (terminal leaf) nodes of the tree are labelled with the end classification, the proportion of results within that node that are positive, and the percentage of observations from the whole dataset present within that node. For example: 92% of the observations had an avELISA less than 0.046 and were classified as negative by the DT; within these observations, 3% were positive.

**Figure 8 animals-14-01113-f008:**
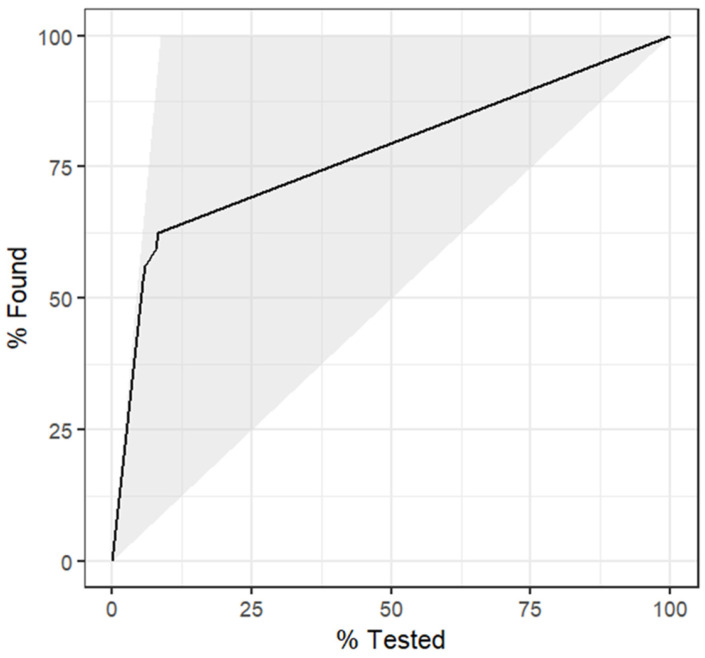
Gain curve for the decision tree using the milk component testing results, demographic data, and Johne’s test results to predict the positive test results (Johne’s milk ELISA). Less than 70% of the positive test results were within the top 25% of the observations with the highest predicted probability of testing positive.

**Figure 9 animals-14-01113-f009:**
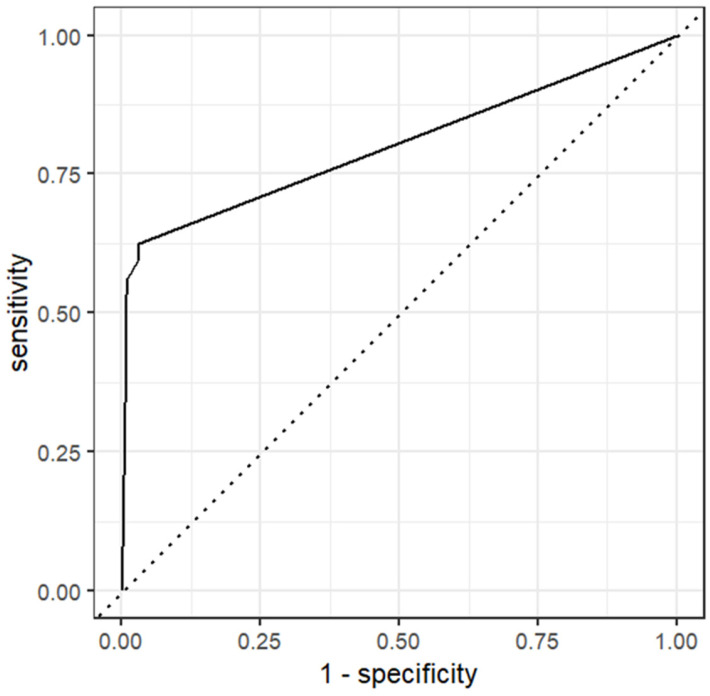
Receiver operating characteristic curve for the decision tree predicting the result of the Johnes milk ELISA with an area under the curve of 0.803.

**Figure 10 animals-14-01113-f010:**
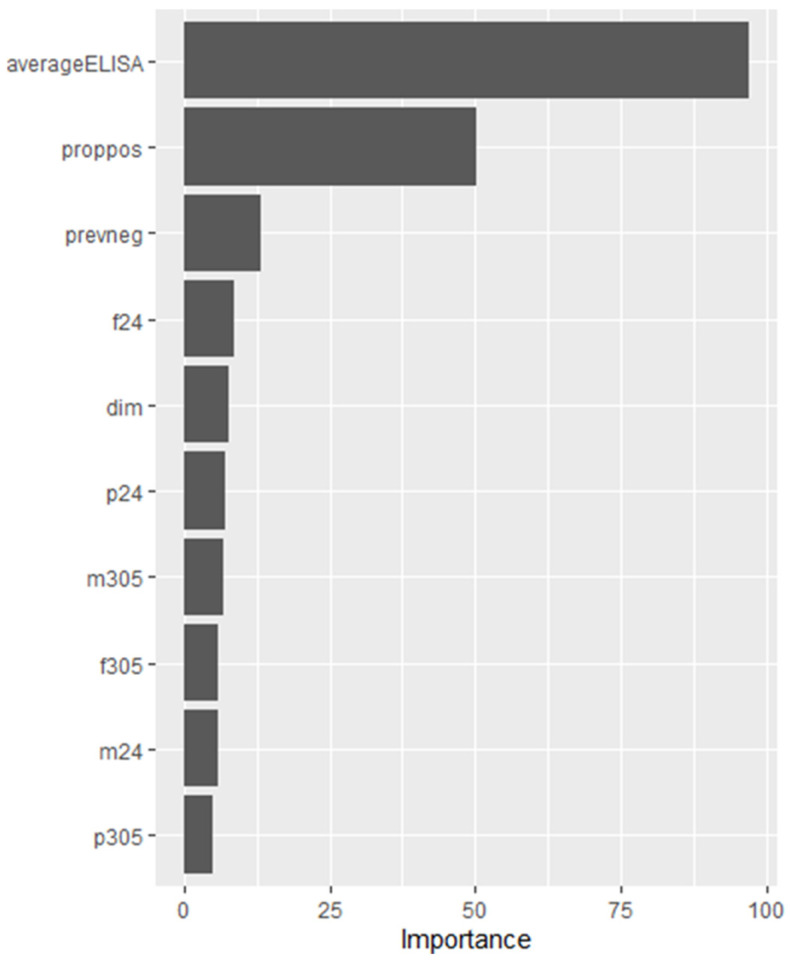
Variable importance plot of the top 10 variables for random forest model 4. m24, f24, and p24: 24 h milk, fat, and protein yields, respectively; m305, f305, and p305: the predicted 305-day milk, fat, and protein yields, respectively; dim: days in milk; prevneg: number of previous Johne’s tests that were negative; proppos: proportion of previous Johne’s tests that were positive; and averageELISA: the average ELISA optical density of previous Johne’s test. The values for variable importance were calculated using the Gini index.

**Table 1 animals-14-01113-t001:** Data subsets used for the training and testing of random forest models used to predict results of Johne’s milk ELISA.

Dataset	Number of Observations	Number of Cows ^1^	Outcome (Levels)	# of Predictors	Model Performance
Subset 1	4008	1059	Interpretation (negative, suspect, positive, high)	10	κ: 0.054, AUC: 0.514
Subset 2	2949	895	Interpretation (negative, suspect, positive, high)	18	κ: 0.518, AUC: 0.763
Subset 3	4008	1059	Interpretation (negative, positive)	10	κ: 0.038, AUC: 0.372
Subset 4	2949	895	Interpretation (negative, positive)	16	κ: 0.626, AUC: 0.915

^1^ The number of cows represented in each data subset. The data used for each of the four subsets contained the Johne’s milk ELISA results from cows with repeat Johne’s tests from eight dairy farms. Subsets 1 and 3 utilized the milk component and demographic data as predictors; subsets 2 and 4 utilized the milk component, demographic, and previous Johne’s test results as predictors. κ: cohen’s kappa; and AUC: area under the curve for receiver operating characteristic curve.

**Table 2 animals-14-01113-t002:** Comparison of the training and test dataset splits for the four data subsets of 4008 Johne’s test results on 1059 Ontario dairy cows used in random forest models to predict the results of Johne’s milk ELISA.

		Training Dataset				Test Dataset		
Data Subset	1	2	3	4	1	2	3	4
Number of observations	3006	2211	3006	2211	1002	738	1002	738
Proportion of negative results	0.92	0.9	0.93	0.92	0.91	0.91	0.92	0.91
Proportion of suspect results	0.01	0.01	NA ^1^	NA^1^	0.01	0.01	NA ^1^	NA ^1^
Proportion of positive results	0.06	0.07	0.07	0.08	0.07	0.07	0.08	0.09
Proportion of high results	0.01	0.01	NA ^1^	NA ^1^	0.01	0.01	NA ^1^	NA ^1^

^1^ Training and test datasets 3 and 4 had a dichotomous outcome and therefore the suspect and high categories were not applicable in the split. The splits were stratified by outcome to ensure equal proportions of outcome classes between the training and testing datasets.

**Table 3 animals-14-01113-t003:** Variable importance values for the top 10 variables for the four final random forest models used to predict the result of Johne’s milk ELISA.

Model 1									
(mtry = 5, min_n = 2, and trees = 1)									
m305	dim	m24	f305	p305	f24	sccls	p24	lact	breed
72.77	66.52	62.21	58.38	52.01	48.93	43.69	41.22	10.04	5.71
Model 2									
(mtry = 18, min_n = 21, and trees = 1000)									
proppos	avELISA	f24	p24	m305	dim	m24	f305	p305	sccls
82.27	78.94	16.26	15.42	14.04	12.03	11.55	11.54	11.12	9.43
Model 3									
(mtry = 1, min_n = 21, and trees = 1)									
f305	dim	p24	m305	sccls	p305	lact	m24	f24	breed
31.36	28.37	16.92	15.82	15.72	13.88	12.71	11.64	9.69	2.92
Model 4									
(mtry = 8, min_n = 2, and trees = 2000)									
avELISA	proppos	prevneg	f24	dim	p24	m305	f305	m24	p305
97.15	50.25	13.15	8.47	7.80	7.19	6.82	5.96	5.94	4.83

The values for variable importance were calculated using the Gini index. Models 1 and 3 used the data from milk components testing only. Models 2 and 4 used the milk component testing results, as well as the Johne’s test results. Models 1 and 2 made predictions on a multiclass outcome (negative, suspect, positive, or high) while models 3 and 4 predicted a binary outcome (negative or positive). Mtry = number of variables sampled at each split; trees = number of decision trees; min_n = the minimum number of data points in a node that is required for the node to be split further; m24, f24, and p24: 24 h milk, fat, and protein yields, respectively; m305, f305, and p305: the predicted 305-day milk, fat, and protein yields, respectively; dim: days in milk; lact: lactation number; breed: cow breed; sccls: somatic cell count linear score; prevneg: number of previous Johne’s tests that were negative; proppos: proportion of previous Johne’s tests that were positive; and avELISA: the average ELISA optical density of previous Johne’s test.

**Table 4 animals-14-01113-t004:** Kappa measures for various combinations of hyperparameter settings applied to model 4 for predicting the results of the Johne’s milk ELISA.

Model	Mtry	Trees	Min_n	Kappa	Out-of-Bag Error
4-A ^1^	8	2000	2	0.727	0.042
4-B	16	2000	2	0.724	0.043
4-C	8	2000	21	0.737	0.041
4-D	8	1000	2	0.737	0.042
4-E	8	1000	40	0.717	0.041
4-F	16	1000	40	0.727	0.042

^1^ The hyperparameters of model 4-A were determined to be the model parameters with the best kappa score according to the tuning grid using a 5-fold cross-validation of the training dataset. Mtry = number of variables sampled at each split, trees = number of decision trees, and min_n = the minimum number of data points in a node that is required for the node to be split further.

## Data Availability

The data are contained within the article.

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
