# Peer review of "Comparison of Machine Learning Tree-Based Algorithms to Predict Future Paratuberculosis ELISA Results Using Repeat Milk Tests"

_animals, 2024, doi:10.3390/ani14071113_

Round 1

Reviewer 1 Report

Comments and Suggestions for Authors

This study explores the application of tree-based machine learning algorithms, specifically decision trees and random forest, to analyze repeat milk testing data from 1,197 Canadian dairy cows for the diagnosis and control of Johne’s disease. Johne’s disease is a chronic, contagious bacterial infection affecting the intestines of ruminants, particularly cattle. The study aimed to predict future Johne’s test results using these algorithms. Random forest models, which utilized milk component testing results alongside past Johne’s results, showed promising predictive performance for a future Johne’s ELISA result, which is a commonly used diagnostic test for the disease. The final random forest model achieved a kappa score of 0.626, a receiver operating characteristic area under the curve (ROC AUC) of 0.915, a sensitivity of 72%, and a specificity of 98%. Additionally, the positive predictive value was 0.81, and the negative predictive value was 0.97. Decision tree models, while providing an interpretable alternative to random forest algorithms, exhibited a slight decrease in model sensitivity compared to random forest models. Overall, the results of this study suggest a promising avenue for future targeted Johne’s testing schemes in dairy cattle. However, further research is needed to validate these techniques in real-world settings and explore their integration into prevention and control programs for Johne’s disease in livestock.

The authors evaluated tree-based machine learning models, specifically models 2 and 4, which accurately predicted 504 Johne's disease (JD) milk test results. However, the uncertainty of JD test results due to poor diagnostic performance complicates management decisions based on these results. The dataset included 105 animals whose test results changed over time, highlighting the challenge of confidently selecting animals for removal from the herd to limit the spread of JD. Model 1 and model 3 showed evidence of overfitting, while there was no evidence of overfitting in model 4. Efforts were made to minimize overfitting by using cross-fold validation, bootstrap aggregation, and limiting the number of variables tried at each split. Model 4, which included concurrent milk component testing results, cow demographic data, historical JD testing of the cow, and JD positivity rate of the herd, showed the highest accuracy in predictive probability. This model could aid producers in targeting cows for follow-up testing, potentially replacing whole herd testing by focusing on animals most likely to test positive.

The most influential variables in model 4 were average ELISA optical density of previous tests and the proportion of previous tests that were positive. These variables indicate the importance of consecutive positive results in predicting future positive ELISA results, aligning with previous research findings.

Variables such as milk protein and fat concentrations, milk yield, days in milk (DIM), and somatic cell count (SCC) were among the top predictors across the models. Lactation number and herd positivity rates were also important predictors. However, interpreting the significance of these variables beyond their importance in prediction is challenging with random forest (RF) algorithms, known as 'black box' algorithms.

Decision tree (DT) models, while providing interpretability, sacrificed prediction accuracy compared to RF models. DT models are generally less accurate than RF models due to their simpler structure.

The study acknowledged limitations such as the imperfect nature of JD tests, potential model overfitting, and the challenge of model interpretation with RF algorithms. Future research should focus on addressing these limitations and collecting more detailed data to improve model performance.

Additionally, the study emphasized the importance of predictor selection in model development, suggesting that predictors should be chosen based on prior knowledge to maximize model performance. Researchers should consider reducing unexplained variability in farm- and cow-level data to improve model performance.

Overall, the study highlights the potential of machine learning models in JD diagnosis and management but also underscores the need for further research to address limitations and improve model accuracy and interpretability.

Author Response

The authors appreciate the thoughtful feedback from reviewer 1. We do hope that this work can provide the foundation for future work/research on this type of analysis hopefully providing improved disease surveillance and disease control initiatives.

Reviewer 2 Report

Comments and Suggestions for Authors

Dear Authors,

although I found merit in your study, I have some concerns, especially regarding the materials and methods section. Please see specific comments/suggestions reported below.

L.76-78: For other studies addressing disease detection in veterinary and animal science (e.g., in the livestock sector) by applying machine learning methods see for example: 

Cockburn, M. Application and prospective discussion of machine learning for the management of dairy farms. Animals 10(9), 1690 (2020). 

L. 80-82: In the dairy sector, data recorded by automatic milk recording systems, as well as DHI records, have been used in different machine learning studies, e.g. to predict mastitis or lameness. Please see for example:

- Ebrahimi, M. et al. Comprehensive analysis of machine learning models for prediction of sub-clinical mastitis: Deep Learning and Gradient-Boosted Trees outperform other models. Comput. Biol. Med. 114, 103456 (2019). 

- Hyde, R. M. et al. Automated prediction of mastitis infection patterns in dairy herds using machine learning. Sci. Rep. 10, 4289 (2020). 

- Warner, D. et al. A machine learning based decision aid for lameness in dairy herds using farm-based records. Comput. Electron. Agric. 169, 105193 (2020).

 L. 193-195: Which was the rationale of separating “positive” results from “high” (highly positive)?

L 195: How many herds were involved in the study? If 8 as reported in Figure 1, please add this information also in the text. Can authors also provide some information about cows (e.g. breed, average milk production, scc…)?

L 215-218: Did authors consider running a repeatability model (e.g., linear mixed model)? A linear model should also be used as benchmark analysis to compare the predictive ability of tree-based models.

L 240: How did the authors deal with the presence of variables with missing data in the analysis? Were missing values imputed? How many variables were excluded as more than 80% of their data was missing?

L 243-250: Did authors consider also applying a feature selection instead of removing highly correlated features with similar biological measurements?

L 271-275 Did authors consider to split datasets 2 and 4, which include animals’ repeated measures, according to animals’ ID, so that different cows will be included either in training or testing datasets? Please see for example:

Bobbo, T. et al. Exploiting machine learning methods with monthly routine milk recording data and climatic information to predict subclinical mastitis in Italian Mediterranean buffaloes.  J. Dairy Sci. 106(3), 1942–1952 (2023)

L 280-289: Given the unbalanced classes of the outcome, did authors consider applying data augmentation strategies (e.g. upsampling, ROSE, SMOTE)?

Table 3: Which is the scale of variable importance values?

L 596-599: See previous comments about repeatability models or the possible split of repeated data by cow ID to avoid overfitting due to data leakage.

L 600-606: Did authors consider running a linear model including OD as dependent variable and the commercial test as independent variable to adjust for the “test effect”. The residuals of this model (adjusted OD) could then be used to define the positive/negative result or itself as an outcome to be predicted.

Author Response

Thank you for your thorough and thoughtful feedback.

Reviewer 3 Report

Comments and Suggestions for Authors

The manuscript "Comparison of machine learning tree-based algorithms to predict future paratuberculosis ELISA results using repeat milk tests" provides a comprehensive examination of the application of tree-based machine learning algorithms to predict future Johne's disease test results in dairy cattle based on repeated milk testing data. Here is a detailed review based on the requested criteria:

Novelty

The study presents a novel approach by applying tree-based machine learning models, specifically decision to predict future paratuberculosis (Johne's disease) test results in dairy cows using repeated milk test data. This application of machine learning in predicting Johne's disease outcomes from milk test data, particularly the use of random forest models for dichotomous outcome prediction (positive vs. negative), demonstrates a novel contribution to veterinary science, particularly in the context of disease control in dairy herds.

Connection of Introduction with Content

The introduction clearly outlines the problem statement, emphasizing the challenges in diagnosing and controlling Johne's disease in dairy herds. It successfully establishes the relevance of machine learning, particularly tree-based algorithms, as a promising solution for enhancing predictive capabilities in disease management. The transition from the problem statement to the proposed solution is logical, setting a strong foundation for the study's objectives.

One suggestion for the introduction section is that it could be shorter. 

Methodology

The methodology is well-documented, detailing data preparation, model building, and the rationale behind choosing specific machine learning algorithms. The process of creating training and testing datasets, handling missing data, and optimizing model hyperparameters is thoroughly explained, ensuring replicability. The choice of metrics for model evaluation, such as kappa and roc AUC, is appropriate for the study's goals.

Presentation of Results

The results are presented clearly, with detailed descriptions of the performance of the four model variants tested. The inclusion of performance metrics, importance of variables, and gain curves for each model variant facilitates a comprehensive understanding of the models' predictive capabilities and their potential practical application. The comparison between decision tree and random forest models in terms of accuracy, interpretability, and model performance offers valuable insights into the trade-offs involved in choosing a modeling approach.

Link Between Results and Discussions

The discussion effectively links the results to broader implications for Johne's disease management in dairy herds. It addresses the practical aspects of implementing machine learning models in disease prediction and control strategies, highlighting the advantages of using random forest models for their higher accuracy and decision trees for their interpretability. The discussion on the challenges of model overfitting, data quality, and the need for larger datasets to improve model performance provides a critical reflection on the study's findings and future research directions.

Conclusions According to the Results

The conclusions drawn from the study are in line with the results obtained, emphasizing the potential of machine learning models, particularly random forests, to enhance the predictive accuracy of future Johne's disease test outcomes based on milk test data. The acknowledgment of limitations and suggestions for future research, including the importance of standardized testing and the collection of more detailed data, underscores the authors' understanding of the field's current challenges and opportunities.

Overall Assessment

The manuscript provides a significant contribution to the field of veterinary science by demonstrating the potential of machine learning algorithms in predicting Johne's disease outcomes. The study is well-designed, with a clear connection between the introduction, methodology, results, and conclusions. The detailed presentation of the methodology and results, along with a thoughtful discussion, makes this study a valuable reference for researchers and practitioners interested in applying machine learning to disease management in dairy herds.

Author Response

Thank you for your thorough and thoughtful review. The objective of the introduction was to not only provide a foundation for machine learning in paratuberculosis research but to also provide a narrative around the evolution/current state of machine learning in the veterinary space. As such the introduction is written to provide an overview of machine learning while trying to maintain only the details which help center the current study.

Round 2

Reviewer 2 Report

Comments and Suggestions for Authors

L. 193-195: Which was the rationale of separating “positive” results from “high” (highly positive)?

The “highly positive” classification was part of the Ontario Johne’s program, the rationale being to target and cull “highly positive” animals to remove animals shedding high amounts of pathogen.

Kelton, D. F., Von Konigslow, T. E., Perkins, N., Godkin, A., MacNaughton, G., Cantin, R. (2014) Quantifying the cost of removing fecals shedders in a voluntary Johne’s disease control program. In 12th International Colloquium on Paratuberculosis. Pg 120.

R: Please include this information in the manuscript.

L 215-218: Did authors consider running a repeatability model (e.g., linear mixed model)? A linear model should also be used as benchmark analysis to compare the predictive ability of tree-based models.

Unfortunately due to the data from two different commercial tests without identification of which test is used for each result, there was no continuous outcome variable to build the linear model.

R:  Does “without identification of which test is used for each result” means that authors do not know for each record which test was used? The type of test should also be considered as a possible factor influencing the outcome. In addition, logistic regression can be run with binary outcomes.

L 240: How did the authors deal with the presence of variables with missing data in the analysis? Were missing values imputed? How many variables were excluded as more than 80% of their data was missing?

Thank you for the question, we state that "Variables missing more than 3,500 or 80% of their observations were excluded from further analysis. " in L348 we state that the variable mun was excluded due to a large number of missing values.

R:  What about variables with less than 80% of missing? These were included in the analysis and how missing values were considered?

L 271-275 Did authors consider to split datasets 2 and 4, which include animals’ repeated measures, according to animals’ ID, so that different cows will be included either in training or testing datasets? Please see for example:

Bobbo, T. et al. Exploiting machine learning methods with monthly routine milk recording data and climatic information to predict subclinical mastitis in Italian Mediterranean buffaloes. J. Dairy Sci. 106(3), 1942–1952 (2023)

As the dataset was imbalanced the primary focus was to ensure the training and testing had equal representation from the outcomes. The goal of these classifiers was to assess the predictive capacity of the predictor variables with the particular outcome, having an animal committed to only the test or train dataset did not align with this goal.

R:  Given the unbalanced outcome, data stratification should be applied, as authors performed. The suggestion was to perform data stratification by animal ID, so that having different animals in training or testing would prevent data leakage and overfitting.

Author Response

Thank you again for your time and attention in reviewing our manuscript. Please see the attachment.
